



# **Temporal variability of tropospheric ozone and ozone**
# **profiles in Korean Peninsula during the East Asian**
# **summer monsoon: Insights from multiple**
# **measurements and reanalysis datasets**
Juseon Bak[1,*], juseonbak@pusan.ac.kr
Eun-Ji Song[1,a], eunji@pusan.ac.kr
Hyo-Jung Lee[1], hyojung@pusan.ac.kr
Xiong Liu[2], xliu@cfa.harvard.edu
Ja-Ho Koo[3], zach45@yonsei.ac.kr
Joowan Kim[4], joowan@kongju.ac.kr
Wonbae Jeon[1,5], wbjeon@pusan.ac.kr
Jae-Hwan Kim[5], jaekim@pusan.ac.kr
Cheol-Hee Kim[1,5,*], chkim2@pusan.ac.kr
*[1]Institute of Environmental Studies, Pusan National University, Busan, South Korea*
*[2]Smithsonian Astrophysical Observatory (SAO), Center for Astrophysics | Harvard & Smithsonian*
*[3]Department of Atmospheric Sciences, Yonsei University, Seoul, Republic of Korea*
*[4]Department of Atmospheric Sciences, Kongju National University, Kongju, South Korea*
*[5]Department of Atmospheric Sciences, Pusan National University, Busan, South Korea*
*[a]Currently at Department of Environmental Atmospheric Sciences, Pukyong National University, Busan, South*
*Korea*

*Corresponding Author**









**Abstract**


We investigate the temporal variations of the ground-level ozone and balloon-based ozone profiles at
Pohang (36.02ºN, 129.23ºE) in Korean Peninsula. Satellite measurements and chemical reanalysis products
are also intercompared to address their capability of providing a consistent information on the temporal and
vertical variability of atmospheric ozone. Sub-seasonal variations of the summertime lower tropospheric
ozone exhibit a bimodal pattern related to atmospheric weather patterns modulated by the East Asian
monsoon circulation. The peak ozone abundances occur during the pre-summer monsoon with enhanced
ozone formation due to favorable meteorological conditions (dry and sunny). Ozone concentrations reach
its minimum during the summer monsoon and then remerges in autumn before the winter monsoon arrives.
Profile measurements indicates that ground-level ozone is vertically mixed up 400 hPa in summer while
the impact of the summer monsoon on ozone dilution is found up to 600 hPa. Compared to satellite
measurements, reanalysis products largely overestimate ozone abundances in both troposphere and
stratosphere and give inconsistent features of temporal variations. Nadir-viewing measurements from the
Ozone Monitoring Instrument (OMI) slightly underestimate the boundary layer ozone, but well represent
the bimodal peaks of ozone in the lower troposphere and the interannual changes of the lower tropospheric
ozone in August, with higher ozone concentrations during the strong El Niño events and the low ozone
concentrations in during the 2020 La Niña event.

## 1.  Introduction


Ground-level ozone should be reduced due to its adverse effect as a key air pollutant and greenhouse
gas in the troposphere, whereas stratospheric ozone should be protected for life on the Earth due to its
essential role in shielding harmful ultraviolet (UV) rays from the sun. Ozone is not directly emitted to the
atmosphere, but formed through the photolysis of oxygen molecules ($O_2$) by strong UV strikes in the
stratosphere as well as the photochemical process in which the photolysis of nitrogen dioxide ($NO_2$) by the
lights below 420 nm yields ozone in the troposphere.
This photochemical production has been strongly affected by the human activities damaging the
protective layer of the stratosphere with the emission of ozone-depleting substances (e.g., CFCs, Halon,
HCFCs) as well as boosting the ground-level ozone pollution with the emission of ozone precursors (CO,
VOCs, $NO_X$). In addition, the formation and fate of atmospheric ozone is complicatedly interacted with
meteorology and climate variability (Jacob and Winner, 2009; Lu et al., 2019; Zhang and Wang, 2016),


making it difficult to evaluate impacts of the emission control measures on ozone levels (Dufour et al.,
2021). As well, the tropospheric ozone is strongly influenced by either downward transport of stratospheric
air masses or the horizontal transport of polluted air-masses (Langford et al., 2015; Walker et al., 2010).

A monsoon is a seasonal change in atmospheric circulation and precipitation, affecting transport, wet

deposition, and chemical reactions on ozone and its precursors. The regional seasonality of ozone as well
as the latitudinal differences in ozone seasonality were attributed to the Asian monsoon-driven atmospheric
circulation (Tanimoto et al., 2005; Worden et al., 2009). In particular, impacts of the East Asian summer
monsoon (EASM) on spatiotemporal variations of surface-layer ozone concentrations over China have been
comprehensively addressed. For example, Yin et al. (2019) characterized the geographical distribution of
ozone in China, with a bimodal structure of ozone with a summer trough in the southern China whereas a
unimodal cycle in the northern China. Shen et al., (2022) specified the source-receptor relationships of
ozone pollution over the central and eastern China, mainly modulated by the monsoon circulation. Korean
Peninsula is located in the easternmost part of the Asian continent adjacent to the West Pacific where more
than a half of the total rainfall amount is typically concentrated during a short rainy season called Jangma
in summer, largely controlled by the EASM (Ha et al., 2012). The interannual and regional variabilities of
monsoon rainfall patterns over Korean Peninsula have been continuously and extensively established (Choi
et al., 2020; Ha et al., 2012), but rarely connected to impacts on the chemical composition.

The main objective of this paper is to characterize the temporal variability of tropospheric ozone and

ozone profiles, by linking with the meteorological variability largely controlled by the EASM. Ground-
based and balloon-based observations are collected from the Pohang station (36.02ºN, 129.23ºE) as a
reference dataset. The ground measurements are used to interpret the sub-seasonal variability of surface
ozone, while the vertical seasonality of ozone is investigated from ozonesondes. This paper is a preliminary
activity of the Asian Summer Monsoon Chemical and Climate Impact Project (ACCLIP) campaign
(https://www2.acom.ucar.edu/acclip) to investigate the impact of the Asian Summer Monsoon on regional
and global chemistry. The ACCLIP campaign will operate two aircrafts during the period July to August
in 2022 to measure atmospheric compounds through entire troposphere to lower troposphere over East Asia
and the West Pacific. The second objective of this paper is to evaluate whether the chemical reanalysis data
and remote-sensing data could represent a consistent picture of the summer monsoon impact on ozone
profile distribution. This evaluation will give an insight on the data selection used to fill in
the spatiotemporal gaps of the ACCLIP measurements.



## 2. Data descriptions

### 2.1 In-situ measurements

Ozonesondes are balloon- borne instruments capable of measuring the vertical distribution of atmospheric ozone from the surface to balloon burst, usually near 35 km. The electrochemical concentration cell (ECC)-typed sensor is the most widely employed. ECC ozonesondes have an uncertainty of 5 %–10 % and a precision of 3 %–5 % (Smit et al., 2007). In South Korea, only at the Pohang station ECC sondes have been regularly launched every Wednesday in the afternoon (13:30-15:30 LT) since 1995. Ozonesonde measurements are reported in units of partial pressure (mPa) with vertical resolution of about 100 m by the Korea Meteorological Administration (KMA). Bak et al. (2019) demonstrated that Pohang ozonesondes measurements are a stable set of reference profiles for validating satellite products.

Surface in-situ measurements of $O_3$ and $NO_2$ are collected from air quality monitoring networks of the National Institute of Environmental Research (NIER) (AirKorea, http://www.airkorea.or.kr). This network measures hourly air pollutants ($O_3$, $NO_2$, CO, $SO_2$) mixing ratios through the chemiluminescence technology (Kley and Mcfarland, 1980). The KMA operates automatic synoptic observation system (ASOS) at 102 weather stations. The ASOS measurements are provided in five types of time scales (minutely, hourly, daily, monthly, yearly) via the KMA Weather Data Service (https://data.kma.go.kr/). We used daily averages of air temperature, relative humidity, solar irradiance, total precipitation, wind speed, and wind direction.

### 2.2 Satellite measurements

Both OMI and MLS were launched on board of NASA's EOS-Aura spacecraft in July 2004 and still functioning in measuring the Earth's atmospheric composition. The Aura satellite crosses the equator at ~ 1:30 in the afternoon. OMI is a nadir-viewing imaging spectrometer capable of daily, global mapping at relatively high spatial resolution of 13 km × 24-48 km (across × along track). MLS measures microwave thermal emission from the limb of Earth's atmosphere. Compared to OMI, MLS makes measurements at a good vertical resolution (~ 3 km) in the upper atmosphere, but at relatively coarse horizontal resolutions (~165 km along the orbit track). The version 4.2 of the MLS standard ozone product is used in this study, only for the recommended vertical range from 261 to 0.025 hPa (Schwartz et al., 2015). We used OMI ozone profiles retrieved using the PROFOZ version 2 algorithm which is in preparation for reprocessing OMI measurements to release a new version of the OMPROFOZ research product (Liu et al., 2010). This retrieval algorithm consists of wavelength/radiometric calibrations and forward modeling simulations, with





an optimal estimation inversion where a priori knowledge is optimally combined with measurement
information to obtain a better estimate of the state (Rodgers, 2000). The measurement sensitivity inherently
decreases toward the surface, with the increasing dependence of retrievals on the a priori information (Bak
et al., 2013). OMI sensitivity is very low to surface ozone, with its maximum in the free troposphere (~500
hPa) (Shen et al., 2019).
**2.3 Reanalysis data**
The Modern-Era Retrospective Analysis for Research and Applications, version 2 (MERRA-2), is
NASA's latest reanalysis, spanning the satellite observing era from 1980 to the present (Gelaro et al.,
2017). In addition to a standard meteorological analysis, a global $O_3$ field is driven by atmospheric
dynamics and constrained by satellite $O_3$ measurements using the GEOS-5 atmospheric model and the data
assimilation system. Beginning in October 2004, MERRA-2 assimilates total column ozone from OMI and
stratospheric ozone profiles above 215 hPa from MLS. Note that OMI total column ozone is assimilated to
account for the lower sensitivity of MLS measurements in the lower stratosphere, specifically in clouded
scenes.
The CAMS reanalysis is the latest global reanalysis data set of atmospheric composition produced by the
Copernicus Atmosphere Monitoring Service (CAMS), covering the period from 2003 to present (Inness et
al., 2019). Compared to MERRA-2, multiple satellite measurements were assimilated for the CAMS
reanalysis with ECMWF's Integrated Forecasting System. These included total ozone columns from
SCIAMARCY, OMI, and GOME/2 as well as ozone profiles from MIPAS and MLS after 2005.
Both reanalysis data have similar temporal and spatial resolutions. Merra-2 system produces 3-hourly
analyses at 72 sigma-pressure hybrid layers between the surface and 0.01 hPa, with a
horizontal resolution of $0.625° × 0.5°$. The CAMS reanalysis data provide estimates every 3 hours with a
horizontal resolution of $0.75° x 0.75°$. The vertical resolution of model consists of 60 hybrid sigma–pressure
(model) levels from surface to 0.1 hPa. In this study, we used CAMS global reanalysis (EAC4) monthly
averaged fields at 25 pressure levels (1000 hPa to 1 hPa) as well as MERRA-2 monthly mean data at 42
pressure levels (1000 hPa to 1 hPa). Both datasets provide ozone profiles in the unit of mixing ratio.
**3. Results and discussion**
**3.1. Temporal variability of ground-level ozone**

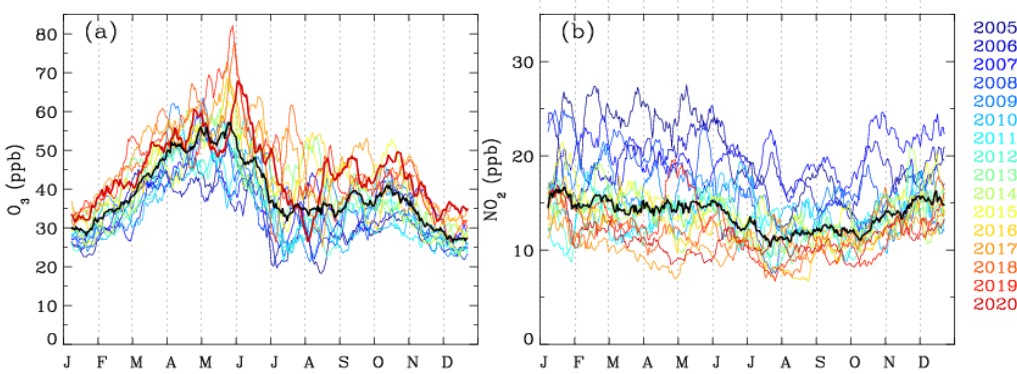


**Figure 1.** (a) Two-week moving averages of daytime ground-level ozone concentrations monitored at 6 sites in
Pohang, with (b) corresponding $NO_2$ concentrations. Different colorings represent each year from 2005 to 2020, while
the black line represents the mean ozone concentrations from all years.


Figure 1 shows both interannual and seasonal changes of daily ground-level concentrations of $O_3$
averaged at six AirKorea sites located within Pohang for 16 years (2005-2020) in comparison with its
primary precursor $NO_2$. Pohang is a major industrial city on South Korea's east coast, with the largest
population of North Gyeongsang Province. In this analysis, hourly measurements in afternoon (1-3 pm
local time) are first averaged for a given calendar day and then smoothed by two-week moving average.
The afternoon $NO_2$ do not change much seasonally. However, the seasonal cycle of ozone is bimodal with
peaks in early-summer and fall. Ozone concentration rapidly increases from ~ 30 ppb in January to primary
peak values of ~ 55 ppb on average during the period of late May to early June. The second peak of ozone
occurs in fall, which is much lower than the major peak.
In wintertime, the annual minimum of ozone concentrations gradually increases by ~ 10 ppb during
last 15 years whereas the annual maximum of summertime ozone rapidly increases from ~ 40 ppb to 80
ppb, in spite of the reduction of $NO_2$ amount by ~ 15 ppb or larger. Both depth and width of the summer
trough are highly variable, likely influenced by the strength and duration of the summer monsoon.



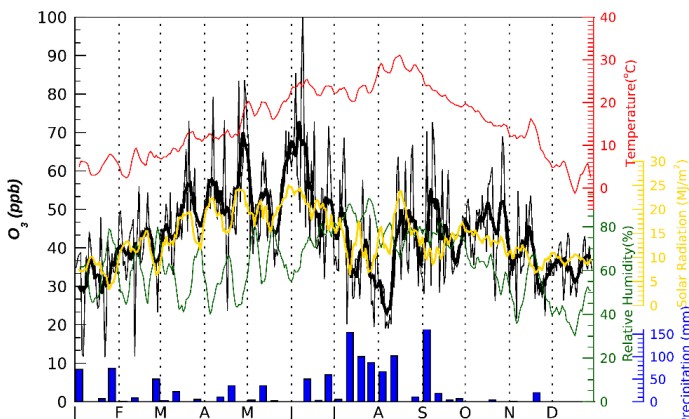


**Figure 2**. (Black) Daily ground-level ozone concentrations where weekly moving averages are applied (thick line) or not (thin line) at Pohang in 2020. The corresponding meteorological factors are overpotted; surface air temperature (red, °C), solar radiation (yellow, MJ/m²), and relative humidity (dark green, %). The bar graph shows the total precipitation (mm) for each weak.

**Table 1**. Same as Figure2, but for correlation coefficients between ozone and meteorological variables, for pre-summer, summer, and post-summer periods, respectively.

|  | pre-summer (Jan-May) | Summer (Jun-Aug) | Post-summer (Sep-Dec) |
|---|---|---|---|
| Solar radiation | 0.91 | 0.74 | 0.51 |
| Air temperature | 0.79 | -0.15 | 0.69 |
| Relative humidity | -0.27 | -0.64 | 0.59 |

In order to avoid smoothing out important features of intra-summer variations in ozone and their association with synoptic weather patterns, daily ozone and meteorological variables are zoomed in 2020 as one-week moving average (Figure 2). The local maximum of ozone concentrations is generally tied to the local warm, dry air and intense solar radiation before the rainy season starts. The correlation between ozone concentrations and meteorological variables is quantitatively compared in Table 1, for summer and post/pre-summer periods, respectively. Solar insolation amounts are directly linked to ozone concentrations over all seasons (r=0.51-0.91). The significant relationship between ozone and air temperature is also identified before and after summer seasons. However, in summer, ozone variations are rarely linked with temperature variations, due to the intense precipitation suppressing ozone formation. Consequently, the local minimum of ozone levels is tied to the local maximum of the relative humidity during the rainy season (r=-0.64). Note that the relative humidity is significantly influenced by air temperature, rather than amount





of water vapor in the pre and post summer periods. Therefore, in the post summer the correlation of ozone
with relative humidity (r=0.59) is likely to arise from the correlation of ozone with air temperature (r=0.51).
The rapid drop of ~ 10 ppb in ozone from the end of July to early August is hardly explained with
meteorological factors mentioned above; the weather becomes warmer with other meteorological variables
(precipitation and solar radiation) being relatively invariant. However, the prevailing wind is characterized
as southwesterlies in early August, exceptionally. Note that the northwesterly winds were dominant in July
and in late August (see. Figure 3). This summer minimum could deepen with the inflow of the poor
ozone airmass originated from the southern sea off the Korean peninsula into inland.

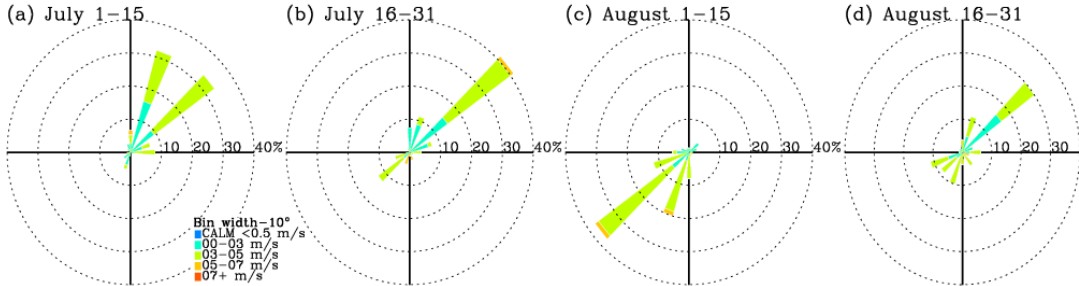


**Figure 3.** Wind roses for individual months from June through September in 2020 at Pohang. Note that hourly
observations in daytime are used to be consistent with data processing done in Figures 1 and 2.

**3.1. Temporal variability of ozone profiles**
To understand the seasonality of ozone profiles, ozonesonde measurements collected at Pohang station
are climatologically averaged for each month and each pressure bin (~ 0.5 km intervals). Ozonesondes
soundings mainly measure ozone in the lower atmosphere below 10hPa while space-based limb soundings
mainly measure ozone in the upper atmosphere above 215hPa. However, both sounding measurements
provide the limited spatiotemporal information. OMI nadir measurements and reanalysis data provide the
daily global maps of ozone profiles. but the reliability of those data products should be assured before using
them to interpret ozone variability and its linkage to the monsoon circulation. As shown in Figure 4a, two
kinds of seasonal patterns are identified with a bimodal structure of layer ozone partial pressures in the
lower troposphere (LT) whereas a unimodal cycle in the upper troposphere and lower stratosphere (UTLS).
The LT ozone concentrations are peaked at June and October with a global minimum in winter as well as
a local summer minimum in late July and early August, which is consistent with surface measurements.



The concentrations of UTLS ozone are relatively higher in March due to the stratospheric intrusion, while
the minimum concentrations appear broadly over the summer and early fall due to the rise of the tropopause,
which is a common feature of ozone in the extratropical UTLS (Gettelman et al., 2011; Rao et al., 2003).
In order to quantify the similarity of seasonal variations, the correlation coefficient is calculated for
temporal ozone changes between each layer and the top/bottom layer. As shown in Fig. 4. b. the seasonality
of ozone at 50 hPa is significantly correlated down to ~ 300 hPa, with the correlation coefficient of larger
than 0.8. In addition, ozone in the boundary layer is significantly correlated with the lower tropospheric
ozone up to 700 hPa (r>0.9) as well as the upper tropospheric ozone up to ~ 300 hPa (r=0.7-0.8). It illustrates
that the 300 hPa could be regarded as a chemical barrier between troposphere and stratosphere at Pohang.

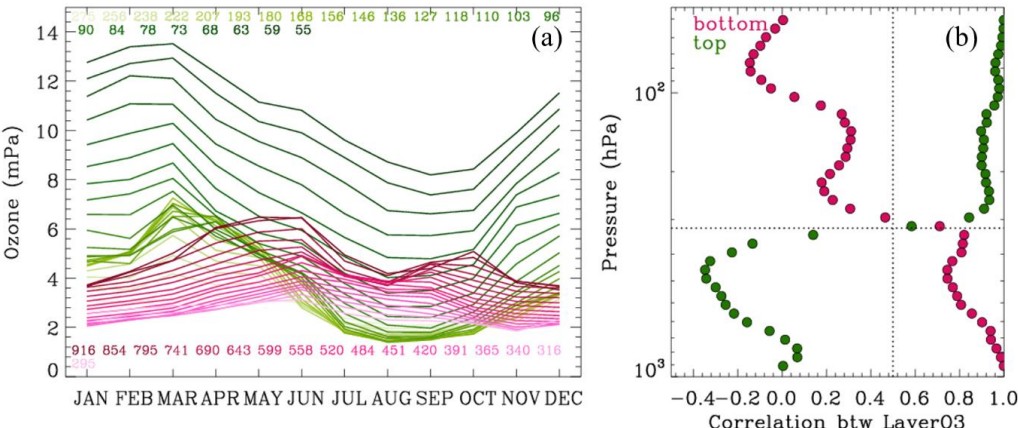


**Figure 4.** (a) Monthly variations of layer ozone partial pressures from ozonesonde soundings obtained from Pohang
during the period of 2005 to 2020. The legend values indicate the midpoint pressure of the layer (hPa). (b) Correlation
coefficients of monthly ozone variations between each layer and bottom layer (916 hPa in red)/top layer (55 hPa in
green).
In Figure 5, monthly averaged ozonesonde profiles are presented for 2020 and compared as a reference
to assess satellite measurements and reanalysis products. This contour map of ozonesondes clearly
illustrates the intrusion depth of the stratospheric air masses down to ~ 300 hPa during spring months (Fig
5a). The mixing depth of ozone that forms near the ground level is also identified, which is bounded up to
~400 hPa in the summer and ~600 hPa in other seasons. The minimum ozone concentration is typically
found just below the thermal tropopause. The August minimum of the lower tropospheric ozone is vertically
extended above ~600 hPa. This air mass is much cleaner compared to the winter ozone concentration over
the lower troposphere. The dominant factor suppressing the ozone formation is a long-lasting summer
precipitation from early July to mid-Aug in 2020 (Fig.2). Southerly wind that blows on the observation site



is relatively strong compared to June and July. Therefore, we could interpret that the inland polluted air
masses are likely to be diluted with the inflows of the maritime clean air masses as mentioned above. In the
lower troposphere, the minor peak of ozone concentrations is also identified in spring, which is not visible
in time-series plots of surface measurements (Fig. 2). The springtime peak is mainly originated by the fair
weather accelerating the formation of ground-level ozone with the wintertime accumulation of ozone and
its processors; it also could be partly attributed by the dynamical processes transporting the ozone-rich airs
from the UTLS and upwind areas. In Figures 5.b-f, OMI, MERRA-2, and CAMS ozone profiles are
qualitatively evaluated with respect to the capability of reproducing the seasonality of ozone profiles  at
this location. The ozone minimum of summer monsoon season is detected from all ozone products, but
much broader than that in ozonesondes due to both the limited time resolution of ozonesonde measurements
and the limited spatial resolution of OMI and reanalysis products. OMI also show a very good agreement
with both ozonesonde in terms of reproducing the boundary layer ozone extending up to free troposphere
and low ozone concentration below the tropopause. In addition, the vertical gradient of ozone enhancement
above the tropopause is consistently reproduced from OMI, ozonesondes, and MLS. The spring ozone peak
near surface is not detectable from OMI measurements due to the limited sensitivity to relatively shallow
boundary layers compared to summer (Shen et al., 2019). In Figure 5.d, OMI a priori profile is also
presented to highlight that the summer minimum is derived from the independent information of OMI
measurements, rather than a priori information. It also illustrates that the summer minimum is a regional
feature of tropospheric ozone seasonality, not represented from the climatological data in which long-term
global measurements are composited as a function of month and latitude.

Both MERRA-2 and CAMS considerably overestimate ozone abundances in both troposphere and

stratosphere in spite of that MLS measurements are commonly employed for assimilating stratospheric
ozone profiles. In MERRA-2, the bimodal peaks (April and October) of the lower tropospheric ozone is
inconsistent with others (early summer, September). We also compare how each ozone product represents
the tropopause against thermally defined tropopause heights using the World Meteorological Organization
(WMO) definition (WMO, 1957). There is no universal method to define the ozonepause height, but
threshold values of 100 to 150 ppb in ozone mixing ratios were used to discriminate stratospheric to
tropospheric air masses (e.g., Hsu et al., 2005;  Prather et al., 2011). In this paper, the 150 ppb value is
selected due to similarities of thermal tropopauses with ozone surfaces of 150 hPa from ozonesonde
measurements. As shown, the ozone surfaces at 150 ppb of reanalysis products are positioned in the free
troposphere due to the overestimation errors. Both ozonesonde and Aura measurements show somewhat



consistency between their ozone and thermal tropopause pressures. In particular, OMI shows the strong
consistency with the fact that retrievals near the tropopause are largely constrained with the a priori state
taken from the tropopause-based ozone profile climatology (Bak et al., 2013).

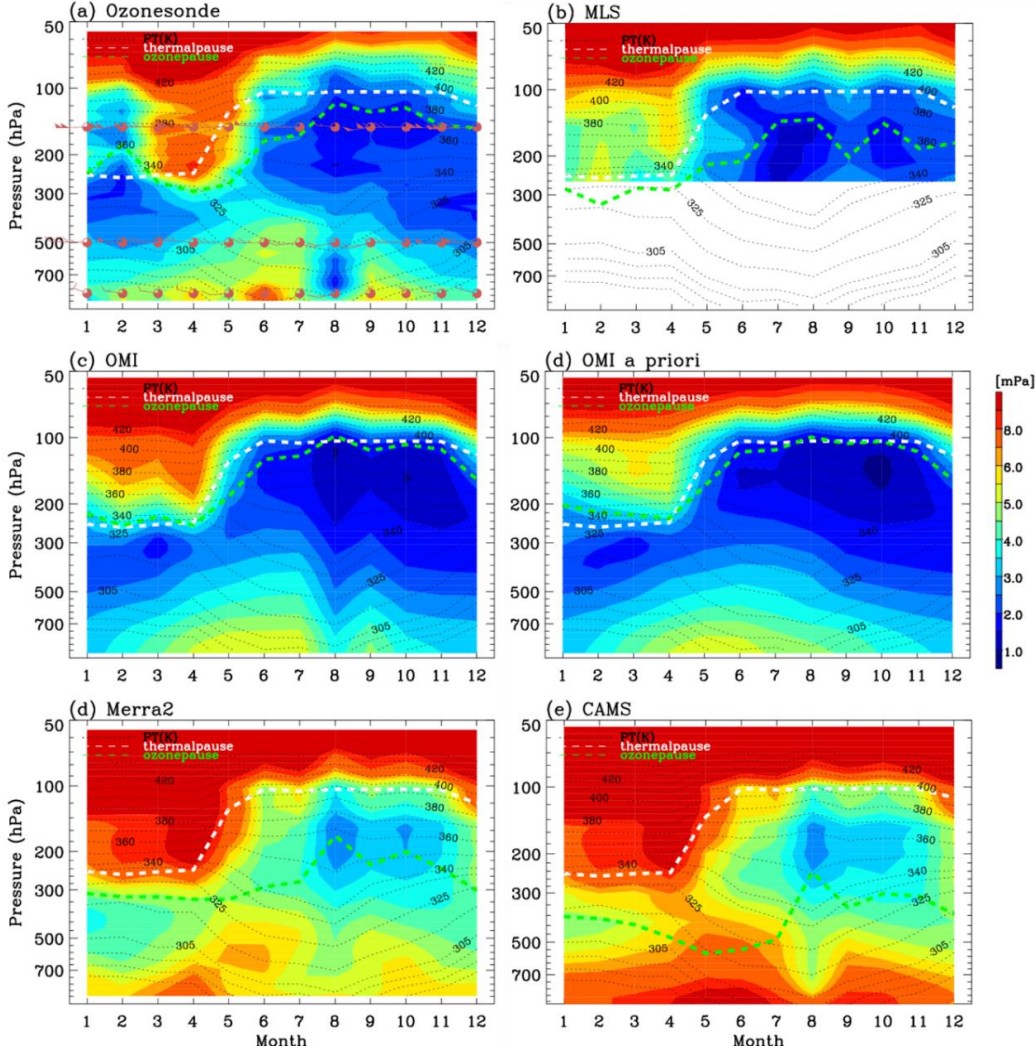

**Figure 5**. Contour plots of monthly ozone profiles in 2020 from (a) ozonesonde, (b) MLS, (c) OMI, (d) OMI a priori, (e) MERRA-2, and (f) CAMS. The meteorological variables are superimposed for wind barbs (red symbols), potential temperatures (black contours), thermal tropopause heights (white lines) using monthly MERRA-2 meteorological data. The ozone value of 150 ppb is plotted with green lines for indicating the chemical transition between troposphere and stratosphere.



### 3.2. Interannual variability of lower tropospheric ozone in summer

In this section, we focus on the ozone changes related to interannual meteorological variabilities, along with the evaluation of different ozone products. In Figure 6, the time-series of mean ozone mixing ratio in the lower troposphere (750-950 hPa) in August are compared. The summer monsoon typically ends in the late July and early August over Korean peninsula and hence the ozone abundance in August is sensitive to the intensity and duration of the monsoon season. OMI and ozonesonde show a similar long-term change, except for much more fluctuations in time-series of ozonesondes due to insufficient samplings (weekly observations) used in monthly averages. A noticeable correlation ($r = \sim -0.52$) exists between wind speeds and ozone mixing ratios (ozonesonde). Low wind speed could enhance the accumulation of ozone precursors and the rate of ozone formation. Accordingly, both ozonesonde and OMI measurements detect higher ozone abundances in August from 2014 to 2017 when the wind speeds are relatively lower. As shown in Figure 7 (a-c), where the monthly meteorological fields at 850 hPa in 2015 are presented from MERRA-2 product, the western North Pacific Subtropical High (WNPSH) was broken in August and hence the weather was likely to be calm and dry over the Korean peninsula. Compared to past few years, the lower amount of ozone is detected in 2020 from ozonesonde measurements. In August 2020, the lower tropospheric southwesterly winds blow from the western North Pacific to Korean Peninsula across the edge of WNPSH as well as the rain belt over Korean Peninsula (Fig. 7. d-f). Therefore, the weather was windy and wet, suppressing ozone formation in August 2020.

MERRA-2 ozone shows no annual variation, before 2020 unlike other ozone measurements and product. CAMS also shows the higher ozone concentrations correlated with wind speeds, but less consistent with ozonesonde measurements compared to OMI. How the El Niño-Southern Oscillation (ENSO) cycle interacts with the East Asian monsoon has been not established. According to the Oceanic Niño Index, the 2015-2016 El Niño event, the warm phase of the ENSO, was one of the strongest events ever recorded, whereas the 2020-2021 La Niña event was also abnormally strong. There was a lot of unprecedented weather events in south Korea during these super El Niño and La Niña periods, such as unprecedented summer rainfalls in 2020 and unprecedented summer heatwaves in 2015-2016 (Yoon et al., 2018). Therefore, we could relate the higher ozone amount in August 2015-2017 and the lower ozone amount in August 2020 to a climatic forcing on the strength and position of WNPSH and hence the East Asian summer climate.

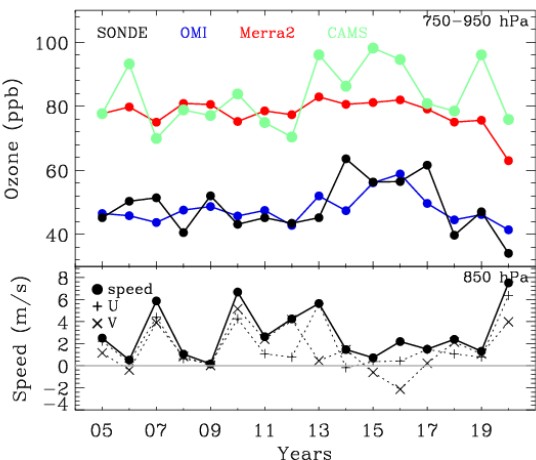

Figure 6. Annual variations of (top) the lower tropospheric ozone (750-950 hPa) in August from various ozone
products, along with (bottom) the wind speeds at 850 hPa.

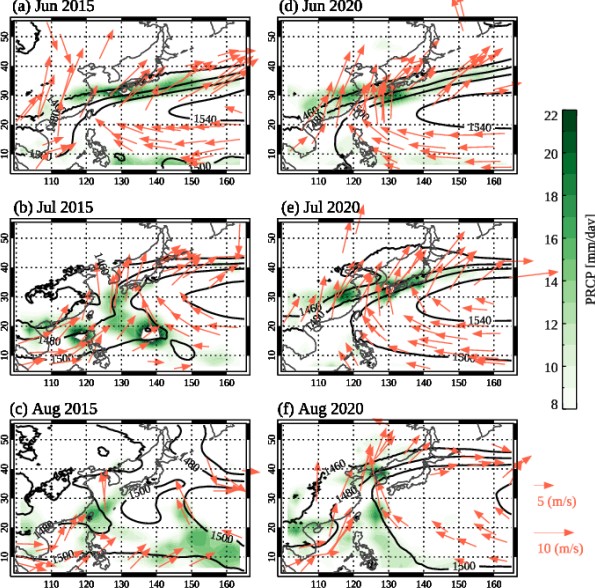

Figure 7. The monthly meteorological fields at 850 hPa for (a-c) 2015 and (d-f) 2020, respectively. The wind vectors
are drawn with the orange arrows. The geopotential heights are superimposed with black lines. The variations of
precipitation are shown with green typed colors, respectively. Note that we use MERRA-2 meteorological variables
except for the precipitation data taken from GPCP Version 2.3 Combined Precipitation Data Set (Adler et al., 2003).





**4 Summary and Conclusions**

In this paper, atmospheric ozone variabilities over Korean peninsula and their linkages to the East Asian summer monsoon are vertically characterized using multiple ozone measurements made by surface observation, balloon-borne ozonesonde, OMI, and MLS. MERRA-2 and CAMS are also integrated in this analysis for the evaluation against ozonesonde. Surface in-situ measurements at six urban sites in Pohang are averaged, while satellite and reanalysis datasets are spatially interpolated onto the Pohang ozonesonde site. Surface measurements clearly show the impact of frequent weather changes (dry and wet) on ozone concentrations in spring. The seasonality of ozone becomes very complicated in late spring to early fall, depending on monsoon strengths and lengths. The peak concentration of ozone occurs in the pre-summer monsoon season (~ 70 ppb) and in the post-summer monsoon season (~50 ppb). During the summer monsoon, ozone concentrations decrease down to ~ 30 ppb, which is even lower than that in the winter when the air temperature and solar insolation is lowest. The vertical structures of ozone concentrations driven by the stratospheric dynamics and synoptic scale tropospheric weather disturbances are characterized from ozonesonde soundings. The stratospheric intrusions actively occur from March to May and modulate the upper tropospheric ozone, down to ~ 300 hPa. We identified ozone enhancements in the boundary layer, extending up to 400 hPa in June. In August the monsoon-induced ozone dilution occurs in the lower troposphere up to ~ 600 hPa. The ozone minimum also occurs just below the tropopause, which is deepest from summer to early fall with the troposphere being extending upward to ~ 100 hPa. Both satellite and reanalysis datasets show the capability of reproducing general features of ozone seasonality such as bimodal peaks in ground-level ozone and spring maximum in the UTLS ozone. However, MERRA-2 and CAMS products significantly overestimates ozone abundances in the UTLS and hence middle tropospheric ozone concentrations exceed 150 ppb which is used as a chemical proxy to distinguish between stratospheric air and tropospheric air. In general, OMI shows a good agreement with ozonesonde measurements with respect to both seasonal tendency and quantitative terms, but slightly underestimates ground-level ozone due to the limited vertical sensitivity. The lower tropospheric ozone in August shows the monsoon-induced interannual variabilities with higher concentrations during the super El Niño and lower concentration during the significant La Niña period, commonly from ozonesonde and OMI measurements. However, MERRA-2 rarely shows long-term changes of August ozone in the lower troposphere. On the other hand, CAMS is annually correlated with ozonesonde measurements, but with the systematic positive biases of ~ 40 ppb. In conclusion, OMI could play a vital role in studying the impact of summer monsoon-derived atmospheric circulation and weather on ozone seasonality. The analysis results





of this study could be a useful reference to the upcoming results from the ACCLIP campaign planned in
the summer of 2022 to gather comprehensive, integrated datasets of two airborne observations (Flight
Operations from S. Korea) and ground/balloon measurements, over the East Asia and Western Pacific.
ACCLIP measurements will provide useful ideas for better understanding the spatiotemporal variation of
ozone in the Korean peninsula in terms of continuous ozone increase near the surface (Yoo et al., 2015),
high ozone in the free troposphere (Crawford et al., 2021), and the relationship between the stratospheric
ozone intrusion and atmospheric circulation (Park et al., 2012).

**Author Contributions** J.B and C.K designed the research; E.S interpreted the reanalysis products
and H.L and W.J contributed on analyzing surface measurements. X.L contributed on OMI ozone profile
retrievals. C.K and JA.K provided oversight and guidance for connecting the weather condition and air
pollutant concentrations. JK and JO.K contributed to the interpretation of the results. J.B lead the writing
of the manuscript; all co-authors contributed to discussion and edited the paper.
**Competing interests**. The authors have no competing interests
**Acknowledgement**
We thank the KMA, NIER, NASA, and Copernicus for providing their measurements and analysis data.
We hope that the 2022 ACCLIP campaign could successfully be processed in South Korea and the research
outcome would be fascinating. We would like to acknowledge the Basic Science Research Program
(2020R1A6A1A03044834 and 2021R1A2C1004984).
*Financial support.* This research has been supported by the Basic Science Research Program through the
National Research Foundation of Korea (NRF) funded by the Ministry of Education (grant
no. 2020R1A6A1A03044834 and 2021R1A2C1004984)
**Data Availability**
Ozonesonde: https://data.kma.go.kr (last access: 16 Jun 2022)
AirKorea: http://www.airkorea.or.kr (last access: 16 Jun 2022)
ASOS: https://data.kma.go.kr (last access: 16 Jun 2022)
OMI ozone profile retrievals: attainable upon request (juseonbak@pusan.ac.kr)
MLS Version 4.2 ozone profile: https://earthdata.nasa.gov (last access: 16 Jun 2022).
MERRA-2 reanalysis data: https://gmao.gsfc.nasa.gov/reanalysis/MERRA-2/ (last access: 16 Jun 2022).
CAMS global reanalysis (EAC4): https://ads.atmosphere.copernicus.eu/ (last access: 16 Jun 2022).





GPCP Version 2.3 Combined Precipitation Data Set: https://psl.noaa.gov/ (last aceess:16 Jun 2022)

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
