# Peer review of "Temporal variability of tropospheric ozone and ozone profiles in Korean Peninsula during the East Asian summer monsoon: Insights from multiple measurements and reanalysis datasets"

_Atmospheric Chemistry and Physics, 2022_

## Referee Comment (RC2)

This manuscript analyzes the temporal variations of the ground-level ozone and ozone profiles measured by ozonesonde in Korean Peninsula. A summer ozone bimodal pattern was found and the effects of the East Asian monsoons on it were assessed. The authors also characterized the temporal variations using satellite measurements and chemical reanalysis products. This study could be a useful reference to understanding the spatiotemporal variation of ozone in the Korean peninsula. In general, I recommend this manuscript for publication after the following comments being addressed.

1. More details for processing of satellite and chemical reanalysis data should be provided, did the authors consider vertical resolution effects between different data?
2. Part of the OMI data is not available due to row anomaly after 2009, how do the authors deal with these gap.
3. The top legends in Figure 4 are not clear, please change the color of the legends.
4. Both "$O_3$" and "ozone" appear in the article, please use one of them throughout the manuscript.

---

## Author Comment (AC1)

**Answer to Referee # 1**

We first thank both reviewers for their thorough comments on our manuscript (MS). We made modifications in accordance with reviewer's comments. **Reviewer's comments in BLACK**, **authors' reply in BLUE**, and corresponding modifications were made by using track change process in the MS.

**Overall Comments:**

This paper is technically valuable to set up the a-priori dataset for satellite retrieval. Also, it is valuable to understand the characteristics of ozone profiles in East Asia. The manuscript is well organized to understand the main purpose of this study, although some typo-error and abbreviation expression are missed. However, some improvements are required before the publication.➔ We appreciate the useful comments, and tried to modify or remove, in accordance with reviewer's suggestion.

**Specific Comments:**

**(C1)** L49-58: In the beginning of the Introduction, two paragraphs are not directly related. Please change the paragraph order.➔ Thank you for pointing to this. We have now changed the order of first two paragraphs in the introduction, and have duly treated to introduce more naturally, as follows,

*Ozone in the lower troposphere should be reduced due to its adverse effect as a key air pollutant and greenhouse gas, whereas stratospheric ozone should be protected for life on the Earth due to its essential role in shielding harmful ultraviolet (UV) rays from the sun. The human activities damage the protective layer of the stratosphere with emissions of ozone-depleting substances (halogen source gases) as well as cause emissions of tropospheric ozone precursors (nitrogen oxides, volatile organic compounds), which chemically react in the presence of sunlight producing ozone in the troposphere. The photochemical formation and fate of ozone in the troposphere is complicatedly interacted with meteorology and climate variability (Jacob and Winner, 2009; Lu et al., 2019; Zhang and Wang, 2016), making it difficult to evaluate impacts of the emission control measures on ground-level ozone (Dufour et al., 2021). (L49~58) in the revised MS)*

**(C2)** L60-62: Is that mentioned for the tropospheric ozone or surface ozone? Surface ozone is still doubtful to consider the affection by the transport from the stratosphere.➔ For clarification, we added 'tropospheric' and 'in the troposphere' in prior sentence (L53~59) in the revised MS.

**(C3)** L71-76: I think that these sentences are not too directly related to any other former sentences. Is that intended to characteristics of EASM? If so, please re-write and change the sentences to before L63. If you intend the ozone-EASM relation, you need to add the chemical status of model.

→ We spitted this sentence (L71~76 in the original MS) into two paragraphs in the revised MS. In the former paragraph, we addressed that the monsoon is a major atmospheric circulation affecting ozone concentrations and also it causes the regional seasonality of ozone and the latitudinal differences. In the paragraph (L59-70), we described that the impact of the EASM on ozone has been already studied comprehensively over China. In the next new paragraph, we highlighted that the relevant studies have been rarely done over Korean Peninsula (KP) with advantages of doing researches in KP; thus we intended to understand ozone-EASM relation qualitatively, not quantitatively (i.e., model approach) in KP, because KP, as being the exit area of the Northeast Asian continent, is highly likely to more sensitive to the influence of the monsoon than China inland. For this clarification, the indicated sentence has been rewritten in new paragraph (L71-77 in the revised MS).

**(C4.1)** Section 2.1: The ozonesonde and surface in-situ measurements can categorizing the "In-situ". However, the importance of ozonesonde is very high in this manuscript. For this reason, please separate the section to "ECC ozonesonde". And detailed explanation is essential, such as the previous studies and heritages of analysis by using the Pohang ozonesonde datasets and detailed uncertainty and biases of ECC ozonesonde by considering the manufactures. → In accordance with the comment, section 2.1 (In-situ measurements) has been divided into two parts: 'Ground measurement' and 'Ozonesonde measurements', and more details on ozonesonde are addressed in the 'Ozonesonde measurements' session especially data screening/filtering process for this study (Line 93~114 in the revised MS).

**(C4.2)** Furthermore, the original observation data of ozonesonde has large uncertainties and some measurement noise errors due to several reasons. For this reason, many of the studies are additionally processed before using the data. For this reason, the author should be also mentioned how the original observation data would be → We agree with reviewer's point; native measurements need to be processed. As stated in the original MS, Bak et al. (2019) have already verified data uncertainties

through cross-evaluation between satellite and ozonesonde dataset over the East Asia region, and reported that Pohang ozonesondes measurements (which are used in the current study) are a stable set of reference profiles. It should be also advised that Pohang datasets used in the current study have been recently reprocessed for the entire period of this study to improve data quality (Daegeun Shin /shingeun@korea.kr, personal communication). In addition, we screened out ozonesondes data obtained under various abnormal conditions (i.e., data at balloon-bursting pressure level exceeding 200 hPa, abnormally high concentration in the troposphere (> 80 DU), or extremely low concentrations in the stratosphere). For clarification about the data screening process, we added the following sentence at the end of the first paragraph in Section 2.1, in the revised MS

*"To improve the data quality, we screened out sounding measurements at the balloon burst altitudes higher than 200 hPa, and observations of either tropospheric ozone column values above 80 DU or stratospheric ozone column values below100 DU.( L109~114 in the revised MS)*

**(C5)** Section 2.2: How about to consider the low tropospheric sensitivity? Because both MLS and OMI are insensitive to the low tropospheric ozone.➔ As well-known and stated in the MS, backscattered UV measurements such as OMI are inherently insensitive to the boundary layer, but relatively more sensitive to the free troposphere (~500 hPa). Nevertheless, as evaluated by Shen et al. (2019), OMI is recognized to be still useful during summertime, sensing to some extent the spatiotemporal variability of boundary layer ozone when boundary layer ozone is extended up to a certain depth and correlated with surface ozone. In our study, Figure 5 also showed that the boundary layer ozone is generally extended up to ~700 hpa (or residual mixed layer of even above 500 hPa) in summer. It was also well recognized through previous multiple studies that OMI reproduces well the seasonally variability of ozone in the troposphere, independent from A priori ozone information. For these reasons above, we did not consider the low tropospheric sensitivity analysis in the current study.

**(C6)** Section 2.3: From Park et al. (2020), some accuracy and characteristics of reanalysis data were listed. They made a conclusion of CAMS & MERRA-2 characteristics both for stratosphere and troposphere, and the reanalysis data has different accuracy characteristics in troposphere and stratosphere. This different

uncertainty characteristic is hard to identify the characteristics of ozone profiles using reanalysis data. As you consider this kind of characteristics, is the analysis confident based on the reanalysis data? In addition, the accuracy characteristics of CAMS and MERRA-2 is different according to the stratosphere and troposphere. The author has to be consider the vertical dependence of uncertainties for reanalysis data for the analysis.

→ Thanks for this important comment. We understand that data production systems of both CAMS and MERRA-2 assimilate the total columns of ozone and stratospheric ozone profiles measured by space-based instruments. Therefore, caution is recommended in general, especially when we use tropospheric ozone profiles taken from reanalysis products because of the absence/uncertainties of tropospheric measurements; this is unlike that stratospheric ozone measured by MLS and MIPAS are generally of high data quality. When comparing CAMS vs. MERRA-2; and each uses a different atmospheric model and data assimilation methodology, overestimations in the UTLS ozone fields are all found by both two data sets, as shown in Figure 5.d and 5.e.

In this background, again the purpose of this study by inter-comparing multiple ozone products is aiming to evaluate how well the chemical reanalysis data and remote-sensing data represent a consistent picture of the summer monsoon impact on ozone profile distributions. As stated in this study, the results indicated that, although our evaluations offer a basis for a qualitative, (not quantitative), comparison between multiple ozone products, CAMS and Merra2 showed insufficient capabilities of representing the seasonality and long-term variability of the lower tropospheric ozone. This is indirectly implying that a regional chemical model should be needed to assist for the improvement of our understandings on the meteorological and chemical impacts on ozone abundances in more detail over Korean Peninsula and its surrounding areas.

**(C7)** L164-165: Only from Figure 1, we can't clarify this sentence. Please add another results to back-up this sentence.

→ It looks obvious that the monsoonal clouds/precipitations are suppressing the photochemical production, and they are responsible for the summer trough of ozone. Figure 2, shown as a back-up Figure to Figure 1 in our study, also illustrates that the duration of the summer trough of ozone

is clearly found during the summer rainy period. It is more obvious in Figure 6, showing that the summer minimum (that is representing the depth of the summer trough) is shown to be linked with the wind speed (that is representing the monsoon intensity). These are all indicating the strong ozone-EASM relation undoubtedly.

In the revised MS, we highlighted these results by moving the conclusionary sentence (L164-165 in the original MS) to L183~185 (after describing Figure 2) in the revised MS, and added two relevant references (Yang et al., 2014; Zhou et al., 2022) below. In these two papers, interannual variabilities of the EASM and its relationship with atmospheric composition were well examined, supporting this sentence "Both depth and width of the summer trough are highly variable, likely influenced by the strength and duration of the summer monsoon".

Yang, et al. 2014: Impacts of the East Asian summer monsoon on interannual variations of summertime surface-layer ozone concentrations over China, Atmos. Chem. Phys., 14, 6867–6879, https://doi.org/10.5194/acp-14-6867-2014.

Zhou et al. 2022: Summer ozone pollution in China affected by the intensity of Asian monsoon systems, Science of The Total Environment, https://doi.org/10.1016/j.scitotenv.2022.157785

**(C8)** From Figure 2 and Table 1, the correlation between ozone and meteorological variables are shown. However, these statistical results only showed the strong relationships, not determined to "Cause and Effect ➔ Local maximum of ozone concentrations is generally tied to the local warm, dry air and intense solar radiation just before starting rainy season, as stated in the MS. In Table 1, we indicated the correlations between ozone and meteorological factors during the summer monsoon as well as during pre/post summer monsoon. The results showed that the comparison emphasizes the lower (or abnormal) correlation between temperature and ozone during the summer monsoon period, mainly due to the precipitation. In particular, the rapid drop of ozone concentrations for the period from the end of July to early Aug does not directly show any correlations between ozone vs. solar insolation, precipitation, relative humidity, and temperature. We could explain this "cause" by the information on wind frequencies (more stated in C9). In this

regard, we believe that Table 1 can be sufficiently used to characterize the daily ozone variations over the year in association with the meteorological conditions.

**(C9)** Figure 3: Pohang is east coast of Korean Peninsula. It means that the South-west wind is transport from land. I think that the South-west wind is not always "poor-ozone airmass".

→ We also agree on reviewer's point; south-west wind is not always "poor-ozone airmass". However, we have shown that concentrations of surface ozone significantly drop over the period from end of July to early August in Figure 2, which does not offer legitimate explanation from other meteorological factors such as temperature, solar insolation, precipitation, and relative humidity. We therefore presented the corresponding wind direction/ speed (wind roses, Figure 3), showing that the prevailing winds have changed from northeast to southwest in 2020. As a result, it is highly likely that the chemical ozone production has been suppressed during the rainy season (July-August), but the southerlies bring the clean oceanic landmasses into Korean peninsula, lowering the background level of ozone in land.

**(C10)** L210: To explain the tropopause characteristics, is "the rise of the tropopause" also satisfied in Pohang? Please add the reference or additional analysis.

→ Yes, we trust that it is also applicable to Pohang. In the current study, the minimum concentration of UTLS was also measured over the summer and early autumn due to the rise of the tropopause. Thus is recognized to be a common feature of ozone in the extratropical UTLS, as stated in Gettelman et al. (2011) and Rao et al. (2003); that were also cited in the current study.

**(C11)** L217: What is "Chemical barrier"? It is too ambiguous. Also, Figure 4 mentioned the correlation characteristic change in 300 hPa. However, this result is analyzed by the long-term datasets. So, this is not suitable to express the special transport phenomenon for STE.

→ We used "chemical barrier" as a boundary between troposphere and stratosphere; that is indicating the tropopause acting as a barrier to upward transport of air and pollutants. Fig.4 shows that seasonal variation of surface-layer concentrations is found to be well consistent with those of partial column ozone in the troposphere below 300 hPa where chemical constituents

tend to be mixed." We add the short explanative words: chemical barrier "working as a boundary" between two (strato- and tropo-) spheres in the revised MS (L226).

**(C12)** L227-232: To describe the vertical diffuse structure, the monthly averaged value is not suitable. Do you have some specific profile cases to support this sentence?

→ August minimum in ozone is intended to represent the ozone seasonality mainly controlled by the summer monsoon, not intended to describe the vertical diffusivity by L227~232. Regarding the reviewer's point that a single profile would not support this feature, coauthors also agreed with this point and all concurred that, as a future study, further temporal datasets would be needed for discriminating the August minimum with bimodal peaks of ozone concentrations detected in the troposphere up to ~ 600 hPa.

*We thank you for reviewer's thorough comments, and we believe that the comments raised by reviewer has been much more strengthened our MS.*

---

## Author Comment (AC2)

**Answer to Referee # 2**

We first thank both reviewers for their thorough comments on our manuscript (MS). We made modifications in accordance with reviewer's comments. **Reviewer's comments in BLACK**, **authors' reply in BLUE**, and corresponding modifications were made by using track change process in the MS.

**Overall Comments:**

This manuscript analyzes the temporal variations of the ground-level ozone and ozone profiles measured by ozonesonde in Korean Peninsula. A summer ozone bimodal pattern was found and the effects of the East Asian monsoons on it were assessed. The authors also characterized the temporal variations using satellite measurements and chemical reanalysis products. This study could be a useful reference to understanding the spatiotemporal variation of ozone in the Korean peninsula. In general, I recommend this manuscript for publication after the following comments being addressed.

→ Our MS has been revised, by reflecting the reviewer's suggestions and comments, as bellow.

**Specific Comments:**

**(C1)** More details for processing of satellite and chemical reanalysis data should be provided, did the authors consider vertical resolution effects between different data?

→ Thank you for this reviewer's point. In the current study, however, the scope of this paper is to 1) understand the temporal variability of ozone associated with the meteorological variability largely controlled by the summer monsoon over Korean peninsula and 2) to evaluate satellite measurements and chemical reanalysis data for giving an insight on the data selection used to fill in the spatiotemporal gaps of the ACCLIP measurements. In addition, we limited the current study to measurements collected on Pohang station due to the data availability of ozonesondes, because CAMS and Merra2 shows insufficient capabilities of representing the seasonality and long-term variability of ozone; thus the regional chemical model support should be more practical to interpret quantitatively the vertical/temporal ozone variability.

The ACCLIP campaign recently has been launched and carried out during August 2022 to collect airborne measurements over Asian Pacific region, aiming to use in examining the roles of Asian

pollution and monsoon strength in chemistry and climate. During the campaign period, ozonesondes have been also launched at 4 sites in S. Korea: Osan, Anmyeondo, and Yongin beside Pohang station during this campaign, and the Korean modelers are simulating the regional chemistry model (including WRF-chem) coupled with WACCM. In this effort, we could provide details of observed relationship between ozone and meteorology from comprehensive, integrated datasets of ACCLIP in the future study. Therefore, in this characterization-correlated analysis study, we do not think that adjusting vertical resolution effects between different data, which is required for quantifying retrieval quality, is needed.

(C2) Part of the OMI data is not available due to row anomaly after 2009, how do the authors deal with these gap?

→ OMI daily coverage has been seriously damaged since 2009 due to row anomaly. However, a specific location could be still sampled within ~ 2 days by OMI. We applied monthly averages to fill up gaps caused by either row anomaly or cloud screening. We do not expect that our application biases to change the major findings of the present study.

(C3) The top legends in Figure 4 are not clear, please change the color of the legends.

→ We replotted Figure 4, as seen below.

[Figure]

**(C4)** Both "O3" and "ozone" appear in the article, please use one of them throughout the manuscript.

→ Thank you for pointing to this. Throughout the entire MS, "O3" has been denoted as "ozone".

*We appreciate the reviewer's insightful comments that are believed to have much more strengthened our MS.*